# Parameter Efficient Multi-task Fine-tuning by Learning to Transfer Token-wise Prompts

**Muling Wu**[1,2], **Wenhao Liu**[1,2], **Jianhan Xu**[1,2], **Changze Lv**[1,2], **ZiXuan Ling**[1,2]
**Tianlong Li**[1,2], **LongTao Huang**[3], **XiaoQing Zheng**[1,2], **Xuanjing Huang**[1,2]

[1]School of Computer Science, Fudan University, Shanghai, China
[2]Shanghai Key Laboratory of Intelligent Information Processing
[3]Alibaba Group, Zhejiang, China
{mlwu22,whliu22,czlv22,zixuanling21,tlli22}@m.fudan.edu.cn
{jianhanxu20,zhengxq,xjhuang}@fudan.edu.cn

## Abstract

Prompt tuning has been proven to be successful on various tasks by incorporating a small number of trainable parameters while freezing large pre-trained language models (PLMs). However, it is still unsettled how to generate more proper prompts for any individual examples and how to extend prompt tuning to multi-task learning scenarios by leveraging cross-task features. To address these challenges, we propose a token-wise prompt tuning (TPT), in which a bank of finer-grained soft prompt tokens is built for multi-task learning by memory network. The tokens are retrieved from the bank against an input example and assembled to an instance-dependent prompt. Extensive experimental results on 14 datasets demonstrated that the models enhanced by our TPT performed far better than full parameter fine-tuned models and achieved state-of-the-art by tuning only 0.035% parameters.[1]

## 1 Introduction

The architecture of Transformers (Vaswani et al., 2017) has yielded impressive performances on various natural language processing (NLP) tasks and has been widely established as the building block for PLMs. The dominant paradigm is to pre-train on large-scale unlabeled datasets and then fine-tune on task-related datasets (Devlin et al., 2019; Raffel et al., 2019; Radford and Narasimhan, 2018). However, performing full parameter fine-tuning for each task would be prohibitively expensive with the growing model scale. Thus, there has been growing interest in developing parameter-efficient fine-tuning (PEFT) methods (Houlsby et al., 2019; Hu et al., 2021; Ben-Zaken et al., 2021) that strive to achieve results comparable to full parameter fine-tuning with a small number of trainable parameters.

Prompt learning (Lester et al., 2021), as a new effective method of PEFT, can boost the model's

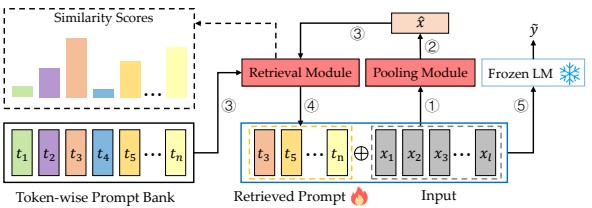

Figure 1: Token-wise prompting solution. An input text of length $l$, $\{x_1, x_2, \cdots, x_l\}$, is sent to a pooling module that produces a feature vector $\hat{x}$ for the input. A retrieval module retrieves the most similar prompt tokens, $\{t_3, t_5, \cdots, t_n\}$, from a prompt bank based on the similarity scores estimated between $\hat{x}$ and $n$ soft prompt tokens $\{t_1, t_2, \cdots, t_n\}$ stored in the bank. The retrieved prompt tokens are assembled to an instance-dependent prompt and concatenated with the original input. The concatenated result is sent to a frozen LM for both inference and training, in which only the retrieved prompt tokens are tuned through error back-propagation.

performance on various tasks by simply adding additional context to the input. Although promising, there are at least two limitations: (a) It overlooks the inherent differences among instances. Even a well-learned prompt might not be suitable for all data instances within a large population, as highlighted by Scao and Rush (2021). (b) It fails to leverage the rich cross-task features, as the learned prompts were exclusively designed for individual tasks, making it difficult for these prompts to be reused or transferred across tasks (Vu et al., 2021).

To overcome the first limitation, we propose a novel approach to automatically generate a more suitable prompt for each input example. Bari et al. (2022) proposed to retrieve non-trainable tokens, also referred to as hard prompt, from the embedding layer of the language model, which can enhance both the training and inference processes of the model. Compared to directly copying the fixed embedding layer as a source of extra information for each example, training an additional embedding layer on the target task can provide more appropri-

---
[1]Our code is available at https://github.com/mlwu22/TPT

ate information. Therefore, we go further on this line by decomposing the trainable soft prompt into finer-grained soft prompt tokens, these tokens constitute the token-wise prompt bank, which can be viewed as a trainable embedding layer. Memory network (Weston et al., 2014) is also used to store, tune and combine these tokens, as illustrated in Figure 1. In contrast to previous methodologies that consider the soft prompt as a single whole, our approach dissects it into fine-grained prompt tokens. This refined breakdown of the prompt widens the scope of search space, facilitating a more exhaustive combination of soft tokens, and ultimately leading to the generation of superior prompts.

To address the second issue, we extend the process of constructing a token-wise prompt bank (i.e., the memory network of token-wise soft prompts) to multi-task learning scenarios. There are many features that can be shared between different tasks, and these features can be learned through multi-task learning (Mahabadi et al., 2021). Additionally, prior work by Vu et al. (2020) demonstrated that performing prompt tuning on intermediate tasks before doing it on the target task can yield even better results. Following their recipe, we first pre-train the token-wise prompt bank across multiple source tasks, then utilize this resulting bank as initialization to train the token-wise prompt bank specifically for the target task.

Finally, we extend our approach, which is called token-wise prompt tuning (TPT), by combining token-wise prompt bank with task-specific prompt tuning, as illustrated in Figure 2. In our approach, all examples within a given task share a task-level prompt, which is generated by way of task-specific prompt tuning. Additionally, for each individual example, an instance-level prompt is retrieved based on the similarity between the input example and tokens in the token-wise prompt bank. These two prompts are concatenated together, incorporating both instance-level and task-level features, as part of the input to facilitate model inference. Moreover, extensive experimental results on 14 datasets demonstrated the effectiveness of our methods.

The contribution of this study can be summarized as follows:

- This study is among the first ones to introduce token-wise prompt tuning by decomposing soft prompts into tokens and constructing a bank of trainable tokens by memory network.
- We extend the token-wise prompt bank to

multi-task learning scenarios, which demonstrates a remarkable boost in transfer learning on both seen and unseen tasks.
- Empirical results on 14 different datasets demonstrate the effectiveness of TPT that outperforms existing prompt-based methods by a significant margin in accuracy and even outperforms the full parameter fine-tuning on both GLUE and SuperGLUE datasets by tuning only 0.035% parameters.

## 2 Related Work

**Task-dependent prompt.** This line of research focuses on enhancing the generation of more effective prompts for specific target tasks.

Specifically, Brown et al. (2020) introduced the utilization of a small set of manually crafted sentences as prompt, typically consisting of task descriptions and relevant examples. The prompt is fed to the frozen model as part of the input, offering the potential to enable the pre-trained model to achieve comparable performance to fine-tuned models, particularly when well-designed.

Auto-prompt (Shin et al., 2020), LM-BFF (Gao et al., 2021), and EFL (Wang et al., 2021a) extend this direction by automating the process of generating discrete prompts. However, optimizing prompts within discrete spaces presents challenges and is likely to be sub-optimal. Prompt tuning (Lester et al., 2021), Prefix tuning (Li and Liang, 2021), and P-tuning (Liu et al., 2021) adopt an alternative strategy by introducing continuous vectors, known as soft prompt, in front of the input sequence. Only these continuous vectors need to be adjusted during training, so the optimization problem in discrete spaces is converted to a continuous optimization task, which can be handled through simple gradient descent.

Moreover, later work has started to consider prompt tuning in the transfer learning scenario. Su et al. (2021) and SPoT (Vu et al., 2021) explore the transferability of prompts learned from different tasks, and address the sensitivity of prompt tuning to initialization through transfer learning. Wang et al. (2023) and PANDA (Zhong et al., 2022) proposed to learn a transfer prompt on the source tasks through knowledge distillation.

**Instance-dependent prompt.** This line of research takes into account the individual characteristics of different examples and generates distinct prompts tailored to each specific example.

In particular, Levine et al. (2022) and IDPG (Wu et al., 2022) generate instance-wise prompts via multi-layer perceptions (MLPs) based on the input encoded by the language model.

Li et al. (2022) and Wang et al. (2021b) maintain a prompt pool to store the prompts learned over source tasks, where each prompt is classified into specific categories and assigned key vectors. The encoded input serves as the query vector, and the target prompts are obtained by weighing the prompts in the prompt pool according to the results of the query and key vector calculations. In addition, ATTEMPT (Asai et al., 2022) calculates weights simply based on the similarity between the input and the prompts learned from the source tasks, without the need for pre-computed clusters.

Unlike these approaches that weighted the soft prompt as a whole, we instead utilize the finer-grained prompt tokens for combination, and only the tokens retrieved according to the input receive the gradient during training. Therefore, a more appropriate prompt can be generated for each example. SPT (Bari et al., 2022) proposed to use retrieved non-trainable hard prompt as a prefix to guide the training of the prompt. In contrast, our method is to retrieve the trainable soft prompt and can be extended to scenarios of multi-task learning to incorporate cross-task features.

## 3 Preliminaries

**Prompt Tuning.** Given a pre-trained LM with parameters $\theta$, and a target task $T_{target}$ with training data $D = \{\mathbf{X}_i, y_i\}_{i=1}^N$, conventional full parameter fine-tuning (FT) seeks to maximize the likelihood of decoding the desired output $y_i$ given input $\mathbf{X}_i$ over training data $D$:

$$\max_{\theta} \sum_{i=1}^N p_\theta(y_i|\mathbf{X}_i) \qquad (1)$$

Unlike FT, prompt tuning freezes the pre-trained language model and only needs to train a very small number of parameters. Specifically, it prepends m randomly initialized vectors, also known as soft prompt $\mathbf{P} = \{p_1, p_2, \cdots, p_m\}$, where $p_i \in \mathbb{R}^d$, before the input $\mathbf{X}_i$, the optimization goal of prompt tuning as follows:

$$\max_{\mathbf{P}} \sum_{i=1}^N p_\theta(y_i|[\mathbf{P}; \mathbf{X}_i]) \qquad (2)$$

where $\theta$ is frozen, and only $P$ is trainable.

**Prompt Transfer.** Transfer learning methods attempt to learn a new target task given a collection of source tasks $T_{source} = \{T_1, T_2, \cdots, T_t\}$, which have been a long-standing way to improve the effectiveness and efficiency of NLP systems(Ruder, 2017). Recent studies such as (Vu et al., 2021; Su et al., 2021) have demonstrated the applicability of transfer learning in the context of prompt tuning, also referred to as prompt transfer. Instead of training the prompt from scratch over target task, these approaches employ the source prompts $\mathbf{P}_{source} = \{\mathbf{P}_1, \mathbf{P}_2, \cdots, \mathbf{P}_t\}$, which are trained according to equation (2) over source tasks. These source prompts $\mathbf{P}_{source}$ can then serve as either initialization vectors or weighted vectors for training the target prompt.

## 4 Method

Our proposed method TPT (As illustrated in Figure 2) consists of two stages: *pre-training token-wise prompt bank* (Section 4.1) and *jointly prompt tuning* (Section 4.2).

TPT pre-trains a token-wise prompt bank that integrates the cross-task features on various source tasks $T_{source} = \{T_1, T_2, \cdots, T_t\}$ and then utilizes this resulting bank as the initial token-wise prompt bank of the next stage to generate instance-level retrieved prompt for each example. In addition, all examples of the target task $T_{target}$ also share the same task-level soft prompt. These two kinds of prompts are concatenated with the input as the final input of the frozen LM and provide both additional instance-level and task-level features for that input, which enhance the model's training and inference processes, leading to improved performance.

### 4.1 Pre-training Token-wise Prompt Bank

We first pre-train a token-wise prompt bank over $t$ high-resource tasks $T_{source}$ via the memory network. The examples from multiple datasets are mixed together, enabling the implicit integration of cross-task features within the learning process of the token-wise prompt bank, thereby endowing it with a powerful capacity for knowledge transfer.

Formally, given the training data $D_{source} = \{D_1, D_2, \cdots, D_k\}$ of source tasks $T_{source}$, one of the input sequence denoted as $\mathbf{X}_i = \{x_1, x_2, \cdots, x_l\} \in \mathbb{R}^{l \times d}$, where $l$ is the input length, $d$ is the dimension of hidden state. The input $\mathbf{X}_i$ and a randomly initialized token-wise prompt bank $\mathbf{B}$ are simultaneously sent to the re-

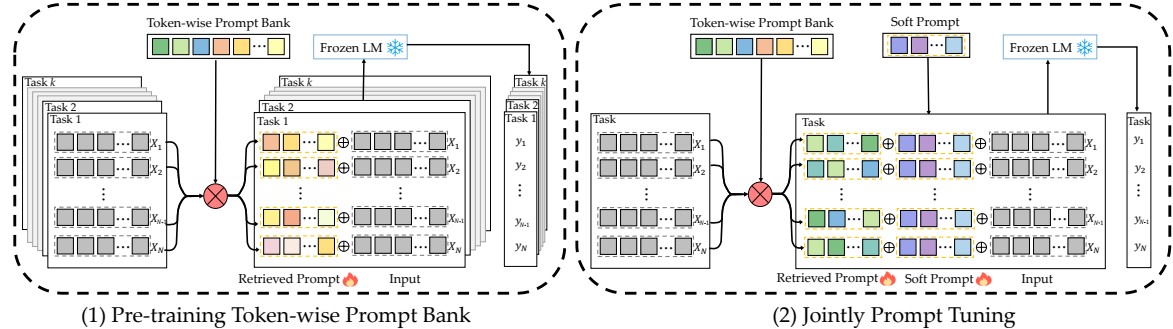

Figure 2: The overall process of TPT. The first step is to pre-train a token-wise prompt bank that absorbs cross-task features on multiple source tasks. The second step utilizes this prompt bank as initialization, transfers the knowledge of the source tasks to the target task and generates a retrieved prompt for each example, and jointly trains with the soft prompt on the target task. Instance-level retrieved prompt and task-level soft prompt provide richer contextual information for input to help the model train and infer better.

trieval module for calculation and an instance-level retrieved prompt $\mathbf{R}_i$ is generated for $\mathbf{X}_i$ according to the similarity results. The retrieved prompt $\mathbf{R}_i$ is prepended in front of the input sequence $\mathbf{X}_i$ to form $[\mathbf{R}_i;\mathbf{X}_i]$, which together serve as the final input of the frozen LM. The training objective is to maximize the likelihood of conditional generating over $D_{source} = \{D_1, D_2, \cdots, D_k\}$, which is:

$$\max_{\mathbf{R}_i \subseteq \mathbf{B}} \sum_{j=1}^{k} \sum_{\mathbf{X}_i \in D_j} p_\theta(\mathbf{y}_i | [\mathbf{R}_i; \mathbf{X}_i]) \qquad (3)$$

### 4.1.1 Soft Prompt Decomposition

Unlike the previous prompt-based method, which weights or combines prompts as a whole (Asai et al., 2022; Li et al., 2022; Wang et al., 2021b), TPT disassembles prompts into finer-grained prompt tokens to complete these operations, enabling a more comprehensive amalgamation of soft tokens, and thereby expanding the range of possible combinations.

For this reason, what is stored in the bank is not the soft prompts trained on the source tasks, but smaller units of prompts, that are, $n$ soft prompt tokens, $\mathbf{B} = \{t_1, t_2, \cdots, t_n\}$, where $t_o \in \mathbb{R}^{n \times d}$. The $n$ soft prompt tokens in the token-wise prompt bank will calculate the similarity scores with the input, and the $k$ tokens with the highest scores will be retrieved and concatenated into the instance-level retrieved prompt $\mathbf{R}_i = \{t_{r_1^i}, t_{r_2^i}, \cdots, t_{r_k^i}\}$, where $r_k^i \in \{1, 2, \cdots, n\}$ indicates that the $k$-th token of the retrieved prompt $\mathbf{R}_i$ generated for the $i$-th example $\mathbf{X}_i$ corresponds to the index of the token stored in the token-wise prompt bank.

During the training process, only the $k$ tokens $\{t_{r_1^i}, t_{r_2^i}, \cdots, t_{r_k^i}\}$ retrieved from the token-wise prompt bank will attain the gradient for adjustment, and the other tokens that have not been retrieved remain untouched.

### 4.1.2 Similarity Score Estimation

The retrieval module controls which tokens to select from the token-wise prompt bank each time an instance-level retrieved prompt is generated by calculating the similarity between the example and prompt tokens in the token-wise prompt bank.

Specifically, the retrieval module will generate the similarity scores $\mathbf{S}_i = \{s_1^i, s_2^i, \cdots, s_n^i\}$ between input $\mathbf{x}_i$ and the n tokens in the bank, where $s_j^i$ denotes the similarity score between $\mathbf{X}_i$ and $j$-th token in the bank. The tokens located at the $k$ indexes $\{r_1^i, r_2^i, \cdots, r_k^i\}$ of that token-wise prompt bank, which possesses the highest similarity scores with the input are retrieved and concatenated into the retrieved prompt of that example, as follows:

$$\{r_1^i, r_2^i, \cdots, r_k^i\} = Index(TopK(\mathbf{S}_i)) \qquad (4)$$

where $TopK()$ function returns the largest $k$ values of the given input, and $Index()$ function returns the subscript of the given input value, which is the index in the bank.

To deal with inputs of various lengths, we apply max-pooling over input $\mathbf{X}_i$ to generate the pooled input $\hat{x}_i \in \mathbf{R}^d$ with the same length of soft prompt tokens $t_j$ in the token-wise prompt bank and the generated $\hat{x}_i$ will be fed into the retrieval module to calculate similarity scores between $\hat{x}_i$ and $t_j$ based on their inner product, which is:

$$\hat{x}_i = MaxPooling(\mathbf{X}_i) \qquad (5)$$

$$s_j^i = \langle \hat{x}_i, t_j \rangle \qquad (6)$$

where $\langle , \rangle$ denotes the operation of inner product.

## 4.2 Jointly Prompt Tuning

When jointly prompt tuning, the token-wise prompt bank trained in the first step is utilized as initialization to generate the instance-level retrieved prompt according to the method described in section 4.1. Transfer learning is performed based on it, and the knowledge of the source tasks is transferred to the target task.

In addition, for all examples of the target task, we initialize a task-level soft prompt $\mathbf{P} = \{p_1, p_2, \cdots, p_m\}$, where $\mathbf{P} \in \mathbf{R}^{m \times d}$, which is shared by them. Instance-level retrieved prompt $\mathbf{R}_i$ and task-level soft prompt $\mathbf{P}$ are concatenated in front of the input to form $[\mathbf{R}_i; \mathbf{P}; \mathbf{X}_i]$, and fed into frozen LM together as the contextual information.

During the training process, retrieved prompt $\mathbf{R}_i$ and soft prompt $\mathbf{P}$ adjust simultaneously, and the optimization objective is transformed to maximize the likelihood of decoding the desired output $y_i$ given input $\mathbf{X}_i$ over training data $D = \{\mathbf{X}_i, y_i\}_{i=1}^N$, as follows:

$$\max_{\mathbf{R}_i, \mathbf{P}} \sum_{i=1}^{N} p_\theta(y_i | [\mathbf{R}_i; \mathbf{P}; \mathbf{X}_i]) \tag{7}$$

In contrast to vanilla prompt tuning, which only provides a shared task-level soft prompt $\mathbf{P}$ for all examples of the target task, TPT provides an additional instance-level retrieved prompt $\mathbf{R}_i$ specific to each example $\mathbf{X}_i$ of the target task as a complementary. This instance-level prompt $\mathbf{R}_i$ captures the particular information related to the input $\mathbf{X}_i$, while the task-level prompt $\mathbf{P}$ encompasses the overall information from the training data $D$ to which $\mathbf{X}_i$ belongs. By incorporating these distinct levels of features, a more comprehensive contextual framework is established for $\mathbf{X}_i$, which can facilitate enhanced model training and inference capabilities.

## 5 Experiments

Following the previous prompt-based methods (Lester et al., 2021; Asai et al., 2022), we perform our experiments on 14 different datasets with full-dataset and few-shot settings, and the experimental results show the effectiveness of our TPT in various scenarios.

### 5.1 Datasets and Tasks

The TPT method is divided into the first stage of pre-training on source tasks, and the second stage of task adaptation on the target task. Specifically, we utilize 6 high-resource datasets as source tasks and select 14 tasks from GLUE (Wang et al., 2018), SuperGLUE (Wang et al., 2019) and SciTail(Khot et al., 2018) as target tasks for evaluation.

**Source tasks.** As in (Asai et al., 2022) and (Wang et al., 2023), we use 6 datasets with more than100k annotations as source tasks: MNLI (Williams et al., 2017), QNLI (Demszky et al., 2018), QQP (Wang et al., 2018), SST-2 (Socher et al., 2013), SQuAD (Rajpurkar et al., 2016), and ReCoRD (Zhang et al., 2018).

**Target tasks.** We selected a total of 14 tasks as target tasks for evaluation, among which, 8 tasks are selected from GLUE: MNLI, QQP, QNLI, SST-2, STS-B (Cer et al., 2017), MRPC (Dolan and Brockett, 2005), RTE (Bar-Haim et al., 2006), and COLA (Warstadt et al., 2018), 5 tasks are selected from SuperGLUE: MultiRC (Khashabi et al., 2018), BoolQ (Clark et al., 2019), CB (de Marneffe et al., 2019), WiC (Pilehvar and Camacho-Collados, 2018), and WSC (Levesque et al., 2011), and another dataset SciTail (Khot et al., 2018). SciTail is a scientific entailment dataset, which is chosen for few-shot evaluation.

### 5.2 Models

Following the standard approach in previous prompt-based method (Lester et al., 2021; Asai et al., 2022; Wang et al., 2023), we mainly experiment using the publicly available pre-trained T5-base model with 220M parameters. In our ablation study, we also consider T5-Small (60M) and T5-Large (770M) models.

### 5.3 Baselines

We compare TPT with the following baselines: (1) full parameter fine-tining (FT), where all the model parameters are tuned during adaptation on each downstream task, while other methods solely focus on adjusting the specific components mentioned below. Specifically, (2) prompt tuning (PT) (Lester et al., 2021), where target prompt vectors are initialized by randomly sampled top vocabularies; (3) SPoT (Vu et al., 2021) and ATTEMPT (Asai et al., 2022) initialize target prompts by retrieving or aggregating prompts trained over source tasks; (4) Adapter (Houlsby et al., 2019) and AdapterDrop (Rücklé et al., 2020) insert trainable modules in the middle of the model; (5) BitFit (Ben-Zaken et al., 2021) only needs to adjust the bias term. (6) MPT

| Method | Param | GLUE | | | | | | | | | SuperGLUE | | | | | |
|---|---|---|---|---|---|---|---|---|---|---|---|---|---|---|---|---|
| | | MNLI | QQP | QNLI | SST-2 | STS-B | MRPC | RTE | CoLA | Avg. | Multi | BoolQ | WiC | WSC | CB | Avg. |
| FT | 220M | **86.8** | **91.6** | 93.0 | 94.6 | 89.7 | **90.2** | 71.9 | 61.8 | 84.9 | 72.8 | 81.1 | **70.2** | 59.6 | 85.7 | 73.9 |
| Adapters | 1.9M | 86.5 | 90.2 | 93.2 | 93.8 | 90.7 | 85.3 | 71.9 | **64.0** | 84.5 | **75.9** | **82.5** | 67.1 | 67.3 | 85.7 | 75.7 |
| AdapterDrop | 1.1M | 86.3 | 90.2 | 93.2 | 93.6 | **91.4** | 86.3 | 71.2 | 62.7 | 84.4 | 72.9 | 82.3 | 68.3 | 67.3 | 85.7 | 75.3 |
| BitFit | 280K | 85.3 | 90.1 | 93.0 | 94.2 | 90.9 | 86.8 | 67.6 | 58.2 | 83.3 | 74.5 | 79.6 | 70.0 | 59.6 | 78.6 | 72.5 |
| ATTEMPT | 232K | 84.3 | 90.3 | 93.0 | 93.2 | 89.7 | 85.7 | 73.4 | 57.4 | 83.4 | 74.4 | 78.8 | 66.8 | 53.8 | 78.6 | 70.5 |
| PT | 77K | 81.3 | 89.7 | 92.8 | 90.9 | 89.5 | 68.1 | 54.7 | 10.6 | 72.2 | 58.7 | 61.7 | 48.9 | 51.9 | 67.9 | 57.8 |
| SPoT | 77K | 85.4 | 90.1 | 93.0 | 93.4 | 90.0 | 79.7 | 68.9 | 57.1 | 82.3 | 74 | 77.2 | 67.0 | 50.0 | 46.4 | 62.9 |
| MPT | 77K | 85.9 | 90.3 | 93.1 | 93.8 | 90.4 | 89.1 | 79.4 | 62.4 | 85.6 | 74.8 | 79.6 | 69.0 | 67.3 | 79.8 | 74.1 |
| TPT | 539K | 85.5 | 90.1 | **93.2** | **94.7** | 89.8 | 89.7 | **82.3** | 59.8 | **85.6** | 74.4 | 80.1 | 69.8 | **67.3** | 94.6 | **77.2** |

Table 1: Results on GLUE and SuperGLUE. All of the results are based on T5-base models. The middle of the table shows the results of the prompt-based method, the top of the table shows the results of other PEFT methods, and the bottom of the table is the result of our proposed TPT. For these experiments, we exclude SQuAD and ReCoRD from source prompts inventories for comparison with prior work. We use Pearson Correlation for STS-B, F1 for MultiRC (Multi), and accuracy for other tasks as metrics. "param/task" denotes the number of parameters trained for each task in GLUE.

(Wang et al., 2023) learns target prompt through knowledge distillation.

### 5.4 Implementation Details

To pre-train the token-wise prompt bank, we conduct a 5-epoch training phase on a mixture of 6 high-resource source tasks. For jointly prompt tuning, we reuse the trained token-wise prompt bank to generate the instance-level retrieved prompt and use the prompt trained on the target task or intermediate task to initialize the soft prompt. Unless specified, we use T5-base as our base LMs for TPT and more details in Appendix A.2.

If a dataset does not have public test split with annotations, we use a development set as our test set or split the development set into our development and test sets, following (Davison, 2021).

In few-shot experiments, for each number of shots k, following (Mahabadi et al., 2021; Asai et al., 2022), we randomly sample 3 times from the training set with different random seeds and report the mean performances. In addition, the task-level soft-prompt is initialized randomly or following (Vu et al., 2021) to use the prompt trained on the MNLI dataset as initialization.

### 5.5 Results

We present main results, which are the full-data adaption in Section 5.5.1, few-shot adaption in Section 5.5.2, and parameter efficiency in 5.5.3.

#### 5.5.1 Full-data adaption

Table 1 presents the per-task performance of different methods on all datasets.

As shown in Table 1, TPT has established state-of-the-art (SOTA) performances on the datasets of GLUE and SuperGLUE compared with these

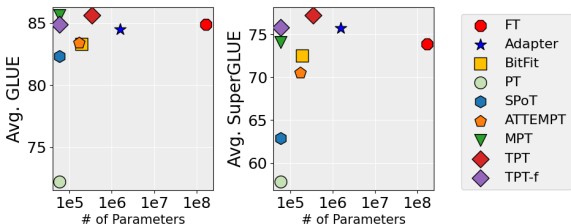

Figure 3: Parameter efficiency on GLUE (left) and SuperGLUE (right). All results are based on T5-base (Raffel et al., 2019).

baselines. Specifically, we achieved SOTA on high-resource datasets: MNLI (93.2%), SST-2 (94.7%), and on low-resource datasets: RTE (82.3%), WSC (67.3%), CB (94.6%), which demonstrate the effectiveness of TPT across various data resource scenarios. Compared to vanilla prompt tuning, TPT obtains a relative improvement of 13.4% on GLUE and 19.4% on SuperGLUE, surpassing the performance of vanilla prompt tuning across all datasets by a large margin. This result further shows that the instance-level retrieved prompt composed of prompt tokens is complementary to the task-level soft prompt generated by prompt tuning, and the supplementary effect is universal. Moreover, TPT outperforms full parameter fine-tuning (FT) on GLUE by 0.7% and SuperGLUE by 4.7%, despite tuning 0.245% as many task-specific parameters.

#### 5.5.2 Few-shot adaption

Following (Mahabadi et al., 2021; Asai et al., 2022; Wang et al., 2023), we conduct few-shot experiments on BoolQ, CB and SciTail, to further verify the effectiveness of TPT under the resource-constrained setup. Table 2 shows the results of our approach and other baselines, which includes

|  | $k$-shot | FT (220M) | AD (1.9M) | PT (77K) | ST (77K) | HF (638K) | ATP (232K) | MPT (77K) | TPT(538k) |
|---|---|---|---|---|---|---|---|---|---|
| BoolQ | 4 | 50.5 | 53.4 | 61.6 | 50.5 | 48.0 | 61.8 | 62.2 | **62.2** |
|  | 16 | 56.5 | 51.4 | 61.9 | 50.6 | 50.2 | 60.0 | 63.3 | **63.5** |
|  | 32 | 58.4 | 54.5 | 61.7 | 61.2 | 58.3 | 65.3 | **68.9** | 67.4 |
| CB | 4 | 57.7 | 51.1 | 53.5 | 71.4 | 60.7 | **82.1** | 73.6 | 78.6 |
|  | 16 | 77.0 | 74.8 | 63.5 | 64.3 | 76.3 | 78.5 | 78.6 | **80.4** |
|  | 32 | 80.0 | 74.8 | 67.8 | 64.3 | 81.4 | 85.7 | 82.1 | **86.3** |
| SciTail | 4 | 79.6 | 79.5 | 57.7 | 69.6 | **82.0** | 80.2 | 80.2 | 81.0 |
|  | 16 | 80.0 | 83.2 | 60.8 | 71.9 | 86.5 | 79.5 | **87.3** | 85.5 |
|  | 32 | 81.9 | 85.0 | 60.2 | 71.9 | 85.8 | 80.2 | **86.3** | 85.2 |

Table 2: Few-shot results ($k$ = 4, 16, 32). FT, AD, PT, ST, HF, and ATP denote Fine-tuning, Adapter, Prompt tuning, SPoT, HyperFormer (Mahabadi et al., 2021), and ATTEMPT.

full parameter fine-tuning, Adapter, prompt tuning, SPoT, HyperFormer, ATTEMPT, and MPT. To be specific, TPT outperforms other methods in certain cases, achieving both SOTA on BoolQ (4, 16-shot) and CB (16, 32-shot). These results clearly indicate that TPT can effectively use cross-task features in source tasks to target tasks in few-shot domain adaptation.

### 5.5.3 Parameter efficiency

Figure 3 compares the performance of different models versus their number of updated parameters on GLUE and SuperGLUE. In addition, TPT-f is a variant of TPT, that is, the parameters of the bank are frozen during the joint training process, and only the parameters corresponding to the task-level soft prompt need to be adjusted.

Specifically, TPT outperforms all other baselines on both GLUE and SuperGLUE with only a small number of parameter adjustments, especially over full-parameter fine-tuning. TPT-f still maintains very high accuracy (y-axis) when adjusting a smaller number of parameters per task (x-axis). TPT-f adjusts as many parameters as vanilla prompt tuning, but the performance on GLUE and SuperGLUE is more than prompt tuning by a large margin, which proves that the TPT and TPT-f have a high degree of parameter effectiveness.

### 5.6 Ablation Study

**Model Scaling.** We empirically analyze how increasing the backbone LM size affects TPT performance. Figure 4 shows the performance of TPT as well as full parameter fine-tuning, Adapter, ATTEMPT, prompt tuning and MPT with three different T5 models (T5-small, T5-base, T5-large). These results show that TPT largely benefits from backbone LM size increase, which is aligned with the finding of (Lester et al., 2021). Furthermore, TPT demonstrates effectiveness across a wide

range of model scales, spanning from 60M to 770M parameters. As the model size increases, TPT exhibits increasingly pronounced advantages. Particularly, when the model size is large, TPT surpasses other baselines on all three datasets.

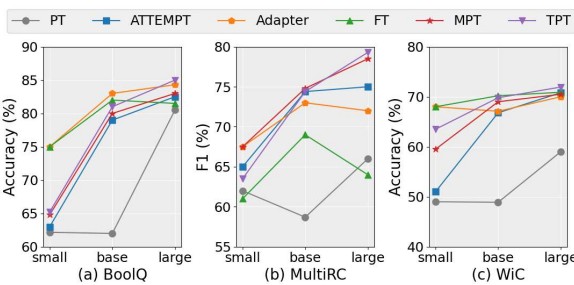

Figure 4: Performance with different backbone LMs.

In addition, we also conducted experiments on a larger scale model (T5-3B), and table 3 shows that our TPT method still has strong competitiveness in the era of large models.[2]

| Method | GLUE |
|---|---|
| PT | 79.7 |
| TPT | 88.1 |

Table 3: Experimental results on large language model (T5-3B).

**Effectiveness of token-wise prompt bank.** We also conduct experiments to assess the effectiveness of solely utilizing the retrieved prompt, abbreviated as RP, generated from the token-wise prompt bank.

Instead of concatenating the instance-level retrieved prompt with the task-level soft prompt and jointly performing prompt tuning, we exclusively prepend the instance-level retrieved prompt, which is composed of tokens retrieved from the token-wise prompt bank, to the input during task adaptation. To be specific, RP-S signifies training the

[2]The large language model (T5-3B) can bring better performance, but the training process is relatively unstable.

prompt bank for the target task from scratch, while RP-M involves pre-training a token-wise prompt bank on multiple source tasks and subsequently employing it as initialization to train the token-wise prompt bank for the target task. In addition, the training method of PR-W is similar to that of RP-S. But unlike RP-S, which decomposes soft prompts into finer grained prompt tokens and then retrieves and adjusts these tokens in the token-wise prompt bank, RP-W treats soft prompts as a whole and then retrieves and adjusts these prompt in the prompt bank.

The results presented in Table 4 reveal that even when using only the retrieved prompt, RP-S outperforms vanilla PT and RP-W by a large margin and RP-M surpasses ATTEMPT by a large margin, which validates that the method of dismantling soft prompts into finer-grained prompt tokens and then combining them can generate a more suitable prompt for each example and also demonstrates the effectiveness of our token-wise prompt bank. Moreover, RP-M performs better than RP-S, which indicates that multi-task learning on source tasks can facilitate a beneficial transfer effect on both seen and unseen target tasks.

| Method | GLUE | SuperGLUE |
|---|---|---|
| PT | 72.2 | 57.8 |
| RP-W | 78.6 | 62.5 |
| RP-S | **83.5** | **67.2** |
| ATTEMPT | 83.4 | 70.5 |
| RP-M | **84.2** | **74.7** |

Table 4: The effectiveness of token-wise prompt bank. "PR" indicates that only the instance-level prompt retrieved from the token-wise prompt bank is used. "-S" means to train the bank from scratch on the target task, "-M" means to perform multi-task learning on multiple source tasks, and then perform transfer learning on the target task to train the bank, and "-W" means to treat the soft prompt as a whole.

**Combination Methods.** We also explore the impact of the two different methods of combining instance-level prompts and task-level prompts on performance: (1) Following the approach of ATTEMPT (Asai et al., 2022), the values of corresponding positions in the two prompts are directly added. (2) Prepending the instance-level prompt in front of the task-level prompt as (Bari et al., 2022).

The results of table 5 show that the second method yields superior performance. This finding suggests that processing the task-level features and instance-level features separately, rather than

directly adding them to an agreement vector, leads to better outcomes.

| Method | GLUE | SuperGLUE |
|---|---|---|
| Addition | 83.4 | 75.2 |
| Concatenation | **85.6** | **77.2** |

Table 5: The impact of different combinations of instance-dependent prompts and task-specific prompts.

**Prompt Initialization.** We explore the impact of soft prompt initialization in the context of the joint prompt tuning process. Our investigation focuses on three distinct initialization strategies: (1) Random Initialization: This approach involves replicating embeddings from the most frequent tokens in the vocabulary. (2) SPoT Initialization: Following the methodology of SPoT, we employ the prompt trained on the MultiNLI dataset as the initialization for the sentence-level classification target task. (3) Target Task Initialization: We utilize the prompt trained specifically for the target task as the initialization. By examining these different strategies, we aim to understand the effects of soft prompt initialization on the overall performance of the joint prompt tuning process.

The results presented in table 6 demonstrate that initializing the task-level soft prompt in three different ways for joint prompt tuning is much better than vanilla prompt tuning, which shows that the soft prompts initialized by these different methods have significantly improved performance on all datasets after being prefixed with the instance-level prompt we proposed for jointly prompt tuning, thus verifying that our proposed instance-level retrieved prompt is complementary to all these different task-level soft prompts.

Furthermore, employing the prompt trained on the target task as the initialization yields the most favorable outcomes, while the randomly initialized prompt exhibits relatively poor results. This observation also indicates the more task-related soft prompt can play a greater role during the joint prompt tuning process.

# 6 Conclusions

In this study, we have introduced TPT, a novel parameter-efficient fine-tuning method designed to address the challenges of generating more suitable prompts for individual examples and extending prompt tuning to multi-task learning scenarios for capturing cross-task features. TPT harnesses the power of a memory network to construct a finer-

| Method | GLUE | SuperGLUE |
|---|---|---|
| Random | 84.4 | 70.7 |
| Intermediate task | 84.7 | 72.9 |
| Target task | **85.5** | **77.2** |

Table 6: The impact of different initialization methods. The results are indicated by "Intermediate task", where the prompts are initialized with those trained on the intermediate task, and by "Target taks" where the prompts are initialized with those tuned on the target task.

grained token-wise prompt bank comprising soft prompt tokens in multi-task learning settings. Our extensive experimental results have demonstrated the effectiveness of TPT. It has demonstrated superior performance over full parameter fine-tuning in specific cases while requiring significantly fewer adjusted parameters. Moreover, these experiments have also verified the compatibility of our method with existing prompt-based approaches, thereby contributing new insights to the field of parameter-efficient fine-tuning.

## Limitations

We have demonstrated the potential of integrating instance-dependent prompts, derived from token-wise prompts, with task-specific prompts to enhance performance. It would be intriguing to examine the feasibility of generating task-specific prompts on-the-fly, leveraging the assembly and retrieval of their token-wise, fine-grained prompts. Additionally, our future research will focus on the creation of a generalized token-wise soft prompts model, which can be applicable across a wide spectrum of NLP tasks, rather than being restricted to a select few.

## Ethics Statement

This work fully comply with the ACL Ethics Policy. All the authors declare that there is no ethical issues in this paper submitted to ACL 2023 for review.

## Acknowledgements

The authors would like to thank the anonymous reviewers for their valuable comments. This work was supported by National Natural Science Foundation of China (No. 62076068), and Shanghai Municipal Science and Technology Project (No. 21511102800).

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

| Method | GLUE | | | | | | | | | SuperGLUE | | | | | |
|---|---|---|---|---|---|---|---|---|---|---|---|---|---|---|---|
| | MNLI | QQP | QNLI | SST-2 | STS-B | MRPC | RTE | CoLA | Avg. | Multi | BoolQ | WiC | WSC | CB | Avg. |
| Random | 85.5 | 90.0 | 93.2 | 94.2 | 89.2 | 88.7 | 80.9 | 54.0 | 84.5 | 72.0 | 78.7 | 69.6 | 63.5 | 69.6 | 70.7 |
| Intermediate task | 84.8 | 90.0 | 93.2 | 93.8 | 89.3 | 89.7 | 80.1 | 56.5 | 84.7 | 73.3 | 79.5 | 67.7 | 63.5 | 80.4 | 72.9 |
| Target task | 84.6 | 90.1 | 93.2 | 94.7 | 89.8 | 89.7 | 82.3 | 59.8 | 85.5 | 74.4 | 80.1 | 69.8 | 67.3 | 94.6 | 77.2 |

Table 7: Experimental results of different initialization methods of task-level soft prompt to perform TPT on GLUE and SuperGLUE.

| Method | GLUE | | | | | | | | | SuperGLUE | | | | | |
|---|---|---|---|---|---|---|---|---|---|---|---|---|---|---|---|
| | MNLI | QQP | QNLI | SST-2 | STS-B | MRPC | RTE | CoLA | Avg. | Multi | BoolQ | WiC | WSC | CB | Avg. |
| PT | 81.3 | 89.7 | 92.8 | 90.9 | 89.5 | 68.1 | 54.7 | 10.6 | 72.2 | 58.7 | 61.7 | 48.9 | 51.9 | 67.9 | 57.8 |
| RP-W | 81.7 | 89.7 | 92.8 | 91.7 | 88.9 | 68.9 | 58.5 | 57.0 | 78.6 | 65.7 | 62.2 | 52.2 | 64.4 | 67.9 | 62.5 |
| RP-S | 82.9 | 89.7 | 92.8 | 92.9 | 89.1 | 88.5 | 72.9 | 58.8 | 83.5 | 74.3 | 62.4 | 52.7 | 64.4 | 82.1 | 67.2 |
| ATTEMPT | 84.3 | 90.3 | 93 | 93.2 | 89.7 | 85.7 | 73.4 | 57.4 | 83.4 | 74.4 | 78.8 | 66.8 | 53.8 | 78.6 | 70.5 |
| RP-M | 83.9 | 90.0 | 93.4 | 93.2 | 89.3 | 88.2 | 78.7 | 56.7 | 84.2 | 73.8 | 78.5 | 67.6 | 64.4 | 89.3 | 74.7 |

Table 8: Experimental results of a single instance-level retrieved prompt on GLUE and SuperGLUE.

| Method | GLUE | | | | | | | | | SuperGLUE | | | | | |
|---|---|---|---|---|---|---|---|---|---|---|---|---|---|---|---|
| | MNLI | QQP | QNLI | SST-2 | STS-B | MRPC | RTE | CoLA | Avg. | Multi | BoolQ | WiC | WSC | CB | Avg. |
| Add | 82.6 | 89.3 | 92.9 | 93.2 | 89.5 | 86.3 | 78.4 | 54.8 | 83.4 | 72.4 | 78.4 | 67.1 | 65.4 | 92.9 | 75.2 |
| Concatenate | 84.6 | 90.1 | 93.2 | 94.7 | 89.8 | 89.7 | 82.3 | 59.8 | 85.5 | 74.4 | 80.1 | 69.8 | 67.3 | 94.6 | 77.2 |

Table 9: Experimental results of TPT on GLUE and SuperGLUE through different prompt combination methods.

| Method | Param | GLUE | | | | | | | | | SuperGLUE | | | | | |
|---|---|---|---|---|---|---|---|---|---|---|---|---|---|---|---|---|
| | | MNLI | QQP | QNLI | SST-2 | STS-B | MRPC | RTE | CoLA | Avg. | Multi | BoolQ | WiC | WSC | CB | Avg. |
| TPT-f | 77K | 84 | 90 | 93.1 | 93.8 | 89.5 | 88.7 | 80.9 | 58.8 | 84.9 | 74.3 | 79.2 | 66.3 | 64.4 | **94.6** | 75.8 |
| TPT | 539K | 84.6 | 90.1 | **93.2** | **94.7** | 89.8 | 89.7 | **82.3** | 59.8 | 85.5 | 74.4 | 80.1 | 69.8 | **67.3** | 94.6 | **77.2** |

Table 10: Experimental results of TPT variant TPT-f on GLUE and SuperGLUE.

# Appendix

# A More details

## A.1 Detailed Results

The following provides more detailed information for the experimental section of this paper.

Table 7 provides a comprehensive breakdown of the outcomes obtained from GLUE and Super-GLUE evaluations, specifically focusing on the task-level soft prompt initialization. This approach involves combining the task-specific soft prompt with the instance-dependent retrieved prompt in order to optimize prompt tuning. The table compares the results for three distinct methods employed in this initialization process.

Table 8 presents a comprehensive analysis of the outcomes obtained from GLUE and Super-GLUE assessments when exclusively relying on the instance-level retrieved prompt as the supplementary context. The aim of this investigation is to evaluate the efficacy of the token-wise prompt bank in generating appropriate prompts for each input example.

Table 9 provides a detailed presentation of the results obtained from GLUE and SuperGLUE eval-uations, specifically focusing on two distinct combination methods employed to combine the instance-dependent prompt and target-specific prompt. The table highlights the outcomes achieved by utilizing these combination approaches.

Finally, Table 10 presents a comprehensive analysis of TPT-f, a variant of TPT, in the context of GLUE and SuperGLUE evaluations. TPT-f effectively reduces the number of adjustable parameters in comparison to TPT, while demonstrating comparable performance in terms of achieved results.

## A.2 Training details

**Hyperparameters.** As used in TPT, we use the prompt length of $m = 100$ for each prompt and use the learning rate of $0.3$ for prompt tuning to train the task-specific prompt and set weight decay to be $1 \times 10^{-5}$. In addition, we also utilize learning rate of $0.3$ for pre-training token-wise prompt bank and jointly prompt tuning and optimize the objective function using Adam (Kingma and Ba, 2014). In particular, we use the learning rate of $0.1$ for SuperGLUE, and Yelp, WinoGrande, SciTail and PAWS multi-task experiments, and $0.3$ for the other experiments. At the same time, we also

try different schedulers and when we train the task-level soft prompt, we choose the constant learning rate of 0.3 and for the other experiments, we also try the linear scheduler.

**Few-shot Adaptation Experiments Details.** Following (Mahabadi et al., 2021), we run few-shot adaptation experiments three times and take the mean of the performance. We cite the performance of the full parameter fine-tuning (FT), Adapter (AD), HyperFormer (HF) from (Mahabadi et al., 2021), prompt tuning (PT), SPoT (ST), ATTEMPT (Asai et al., 2022) (ATP), and MPT (Wang et al., 2023) and random initialize the task-level soft prompt or utilize the prompt trained on MNLI.

**Per-device batch size for** TPT **and prompt tuning**. For T5 small and base, we set per-GPU batch size to be 100 and 32, while for T5-large, we use the batch size of 16.