# OpenReview forum: "Parameter Efficient Multi-task Fine-tuning by Learning to Transfer Token-wise Prompts"
_EMNLP/2023/Conference — EMNLP 2023 Findings_

### Official Review · Reviewer_11wC · 2023-08-02

**Soundness:** 3

**Excitement:**

2: Mediocre: This paper makes marginal contributions (vs non-contemporaneous work), so I would rather not see it in the conference.

**Paper Topic And Main Contributions:**

This paper proposes a method to enhance prompt-based methods for natural language processing tasks. The authors introduce the concept of token-wise prompt banks, which consist of fine-grained prompts that can be combined to generate suitable prompts for specific examples. They demonstrate the effectiveness of their approach through experiments on various NLP tasks, showing improved performance compared to existing methods.

The main focus of the article is to propose a method called token-wise prompt tuning (TPT) to enhance prompt-based methods for natural language processing tasks. The authors aim to address the challenges of generating proper prompts for individual examples and extending prompt tuning to multi-task learning scenarios. They introduce the concept of token-wise prompt banks, which consist of fine-grained prompts that can be retrieved and assembled to create instance-dependent prompts. The article primarily focuses on demonstrating the effectiveness of TPT through experimental evaluations of various NLP tasks.

The token-wise prompt tuning (TPT) approach addresses the limitations of previous methods in two key ways:

1. Individualized prompts: Previous methods often use a single prompt for all examples within a task, overlooking the inherent differences among instances. TPT tackles this limitation by generating instance-dependent prompts. It decomposes the trainable soft prompt into finer-grained soft prompt tokens and retrieves tokens from a token-wise prompt bank based on the similarity between the input example and the tokens in the bank. This allows for the generation of more suitable prompts tailored to each example, enhancing the model's performance.

2. Cross-task feature leveraging: Previous methods design prompts exclusively for individual tasks, making it challenging to reuse or transfer prompts across tasks. TPT extends the construction of token-wise prompt banks to multi-task learning scenarios. It pre-trains the token-wise prompt bank across multiple source tasks, capturing cross-task features. This resulting bank is then utilized as initialization to train the token-wise prompt bank specifically for the target task. By incorporating cross-task features, TPT demonstrates a remarkable boost in transfer learning on both seen and unseen tasks.

**Reasons To Accept:**

1. Novel approach: The concept of token-wise prompt banks is a novel contribution to prompt-based methods in NLP. It provides a more comprehensive and contextually relevant framework for training and inference.
2. Experimental results: The paper presents experimental results on a range of NLP tasks, demonstrating the effectiveness of the proposed method. The results show significant improvements over existing methods, validating the approach.
3. Clear presentation: The paper is well-structured and clearly presents the methodology, experiments, and results. The authors provide sufficient details to understand the proposed approach and its implementation.

**Reasons To Reject:**

1.Implementation Complexity: The method involves several complex components, such as memory networks and a token-wise prompt bank, which might increase the complexity of implementation and tuning.
2.Utilization of Cross-Task Features: While the method attempts to integrate cross-task features through multi-task learning, ensuring effective transfer of these features across different tasks might be challenging.
3.Parameter Efficiency: The method emphasizes achieving performance comparable to full parameter fine-tuning by tuning only 0.035% of parameters. However, this might limit the model's flexibility and ability to adapt to different tasks.
4.Limited comparison: While the paper compares the proposed method with existing approaches, the comparison is limited to a few methods. A more comprehensive comparison with a wider range of state-of-the-art methods would provide a better understanding of the relative performance.

**Reproducibility:**

3: Could reproduce the results with some difficulty. The settings of parameters are underspecified or subjectively determined; the training/evaluation data are not widely available.

**Reviewer Confidence:**

3: Pretty sure, but there's a chance I missed something. Although I have a good feel for this area in general, I did not carefully check the paper's details, e.g., the math, experimental design, or novelty.

---

> ### Author Rebuttal · Authors · 2023-08-27
>
> Thank you for your valuable comments!
>
>  **Q1：The method involves several complex components, such as memory networks and a token-wise prompt bank, which might increase the complexity of implementation and tuning.**
>
> R1: Although our method introduces multiple components, it still only requires adjustments to a very small percentage of parameters (0.035%) compared to full-parameter fine-tuning. Therefore, it should not pose significant implementation challenges.
>
>
>
> **Q2：While the method attempts to integrate cross-task features through multi-task learning, ensuring effective transfer of these features across different tasks might be challenging. **
>
> R2 : In the ablation experiments section, we demonstrated that through multi-task learning, we can achieve excellent transfer learning results on both seen and unseen tasks, as shown in the table below:
>
> |      | GLUE | SuperGLUE |
> | :--: | :--: | :-------: |
> | RP-S | 83.5 |   67.2    |
> | RP-M | 84.2 |   74.7    |
>
>
>
> **Q3：The method emphasizes achieving performance comparable to full parameter fine-tuning by tuning only 0.035% of parameters. However, this might limit the model's flexibility and ability to adapt to different tasks**
>
> R3 : Compared to full-parameter fine-tuning, our method achieves better performance while adjusting fewer parameters. Additionally, when we increase the model size or trainable parameters, we can further enhance our performance.
>
>
>
> **Q4：While the paper compares the proposed method with existing approaches, the comparison is limited to a few methods. A more comprehensive comparison with a wider range of state-of-the-art methods would provide a better understanding of the relative performance.**
>
> R4 : Our work involves comparing with several strong baseline methods in prompt tuning developed previously, such as ATTEMPT (EMNLP2022) and MPT (ICLR 2023).
>
> Furthermore, in the revised version, we will include the following:
>
> 1.  A comparison with SPT, with experimental results as shown below:
>
>    |      | SuperGLUE |
>    | :--: | :-------: |
>    | SPT  |   74.7    |
>    | TPT  |   77.2    |
>
>
>
> 2. Replication of HyperPrompt's experiment, where backbone parameters are frozen and only additional parameters are adjusted, for a fairer comparison with our TPT method

---

### Official Review · Reviewer_viJ7 · 2023-08-08

**Soundness:** 2

**Excitement:**

1: Poor: I cannot identify the contributions of this paper, or I believe the claims are not sufficiently backed up by evidence. I would fight to have it rejected.

**Paper Topic And Main Contributions:**

This paper proposes a method that could retrieve tokens from prompt bank and utilizes them for prompt tuning.

**Reasons To Accept:**

this paper is well written and easy to understand.



**Reasons To Reject:**

1. This paper is similar to previous prompt tuning paper, with the added element of retrieval for prompt initialization. While the approach is straightforward, it doesn't offer significant innovation compared to earlier studies.

2. In this paper, the authors primarily utilized the T5-small and T5-large models, which seem somewhat outdated. It might be beneficial for the authors to explore more recent decoder-only models, like the LLaMa model, for a more current analysis.

3. This paper concentrates solely on classification tasks, including GLUE and SuperGLUE, which can be considered somewhat dated. Generation tasks, by contrast, present more utility and challenges than classification tasks.

In general, this article feels like it's from a few years ago and isn't particularly captivating.



**Reproducibility:**

4: Could mostly reproduce the results, but there may be some variation because of sample variance or minor variations in their interpretation of the protocol or method.

**Reviewer Confidence:**

4: Quite sure. I tried to check the important points carefully. It's unlikely, though conceivable, that I missed something that should affect my ratings.

---

> ### Author Rebuttal · Authors · 2023-08-27
>
> Thank you for your valuable comments!
>
>  **Q1: This paper is similar to previous prompt tuning paper, with the added element of retrieval for prompt initialization. While the approach is straightforward, it doesn't offer significant innovation compared to earlier studies**
>
> R1 : Our work is built upon prompt tuning, with a focus on addressing the limitations of vanilla prompt tuning. Additionally, we introduce a novel approach of disentangling the soft prompt into prompt tokens, which significantly improves performance in a token-wise prompt tuning manner, demonstrating the effectiveness of our method.
>
>
>
> **Q2: In this paper, the authors primarily utilized the T5-small and T5-large models, which seem somewhat outdated. It might be beneficial for the authors to explore more recent decoder-only models, like the LLaMa model, for a more current analysis**
>
> R2: Our work involves comparing with several strong baseline methods in prompt tuning developed previously, such as ATTEMPT (EMNLP2022) and MPT (ICLR 2023). Their method utilizes t5-base as a backbone, and to facilitate a more accurate comparison with their approach, we have also employed t5-base as our backbone.
>
> Furthermore, in the ablation study section, we investigated the impact of model scale. On the three datasets we compared, our approach achieved the best performance when scaled up to T5-large. This demonstrates the potential for our method to achieve even greater performance gains when increasing the model scale.
>
> In the revised version, we will also include experiments with larger-scale backbones, such as llama.
>
>
>
> **Q3: This paper concentrates solely on classification tasks, including GLUE and SuperGLUE, which can be considered somewhat dated. Generation tasks, by contrast, present more utility and challenges than classification tasks**
>
> R2 : Similar to Q2, to facilitate comparisons with the previous baselines, our method was also tested exclusively on classification tasks. Finally, in the revised version, we will evaluate the performance of our TPT method on more complex generation tasks.

---

### Official Review · Reviewer_GwUS · 2023-08-09

**Soundness:** 3

**Excitement:**

3: Ambivalent: It has merits (e.g., it reports state-of-the-art results, the idea is nice), but there are key weaknesses (e.g., it describes incremental work), and it can significantly benefit from another round of revision. However, I won't object to accepting it if my co-reviewers champion it.

**Missing References:**

Not applicable

**Paper Topic And Main Contributions:**

This paper focuses on parameter-efficient fine-tuning for pre-trained language models, specifically, prompt-tuning. The paper proposes a token-wise prompt selection and training method. These prompts are firstly trained on multi-task corpus and then specified to target tasks. The experiment results in this paper verify the effectiveness and efficiency of the proposed method.

**Questions For The Authors:**

Question A: Do you think the method or idea proposed in this paper can directly facilitate or have any implications on the usage and adaption of cutting-edge large language models (e.g. ChatGPT)?

Question B: How long does TPT need for training and inference and how is this time consumption compared to full-parameter tuning?

**Reasons To Accept:**

1. This paper presents token-wise prompt tuning, which is a novel and interesting prompt-tuning approach.
2. The performance shown in the experiment is good.
3. The writing is clear and the structure is well-organized.

**Reasons To Reject:**

1. Experiment: the experiment only covers a model size of 220M. I think a more comprehensive evaluation of a larger range of models with a variance of scales should be included (e.g. T5-large on full GLUE dataset and even larger models), especially for those larger models given the continuous growth of the scales of pre-trained language models.
2. Ablation: the first comparison in Table 3 (PT vs RP-S) seems unfair. To justify the claim "dismantling soft prompts into prompt tokens can generate a more suitable prompt", I think the baseline should be pools of soft prompts with similarity-based retrieval methods rather than vanilla soft-prompt (since the major contribution of the work is dismantling rather than dismantling + retrieval). Moreover, I wonder if ATTEMPT is a multi-task pre-training-based method. If not, then I don't think the second comparison is a fair one.
3. Given the strong ability of currently emerged language models (e.g. ChatGPT) and the accompanying simpler adaption methods (e.g. in-context learning), the proposed method seems outdated.

**Reproducibility:**

4: Could mostly reproduce the results, but there may be some variation because of sample variance or minor variations in their interpretation of the protocol or method.

**Reviewer Confidence:**

3: Pretty sure, but there's a chance I missed something. Although I have a good feel for this area in general, I did not carefully check the paper's details, e.g., the math, experimental design, or novelty.

**Typos Grammar Style And Presentation Improvements:**

line 64: ", these" -> ". These"

line 302: $\mathbf{x}_i$ -> $\mathbf{X}_i$

---

> ### Author Rebuttal · Authors · 2023-08-27
>
> Thank you for your valuable comments!
>
> **Q1: The experiment only covers a model size of 220M. I think a more comprehensive evaluation of a larger range of models with a variance of scales should be included (e.g. T5-large on full GLUE dataset and even larger models), especially for those larger models given the continuous growth of the scales of pre-trained language models.**
>
> R1: Our work involves comparing with several strong baseline methods in prompt tuning developed previously, such as ATTEMPT (EMNLP2022) and MPT (ICLR 2023). Their method utilizes t5-base as a backbone, and to facilitate a more accurate comparison with their approach, we have also employed t5-base as our backbone.
>
> Furthermore, in the ablation study section, we investigated the impact of model scale. On the three datasets we compared, our approach achieved the best performance when scaled up to T5-large. This demonstrates the potential for our method to achieve even greater performance gains when increasing the model scale.
>
> In the revised version, we will also include experiments with larger-scale backbones, such as T5-3B or llama.
>
>
>
> **Q2: The first comparison in Table 3 (PT vs RP-S) seems unfair. To justify the claim "dismantling soft prompts into prompt tokens can generate a more suitable prompt", I think the baseline should be pools of soft prompts with similarity-based retrieval methods rather than vanilla soft-prompt (since the major contribution of the work is dismantling rather than dismantling + retrieval). Moreover, I wonder if ATTEMPT is a multi-task pre-training-based method. If not, then I don't think the second comparison is a fair one.**
>
> R2 : Well, It's a real fantastic question! Previously, retrieved-based methods were designed to retrieve soft prompts trained over various source tasks.  There hasn't been a baseline that solely retrieves soft prompts trained over the target task. Therefore, we have designed a baseline that retrieves soft prompts in a manner similar to TPT, but exclusively from the prompt pool corresponding to the target task. The experimental results are as follows:
>
> |          | MNLI | QQP  | QNLI | SST2 | STSB | MRPC | RTE  | COLA | AVG  |
> | :------: | :--: | :--: | :--: | :--: | :--: | :--: | :--: | :--: | :--: |
> |    PT    | 81.3 | 89.7 | 92.8 | 90.9 | 89.5 | 68.1 | 54.7 | 10.6 | 72.2 |
> | Retrieve | 81.7 | 89.7 | 92.8 | 91.7 | 88.9 | 68.9 | 58.5 | 57.0 | 78.6 |
> |   RP-S   | 82.9 | 89.7 | 92.8 | 92.9 | 89.1 | 88.5 | 72.9 | 58.8 | 83.5 |
>
> The experimental results demonstrate that retrieval-based methods lead to performance improvements, but they still fall significantly behind our TPT method. Thus, this also validates our viewpoint: dismantling soft prompts into prompt tokens can generate a more suitable prompt.
>
> Additionally, our approach can be viewed as a variant of ATTEMPT, as both involve 'pre-training' on the same source tasks and then adapting to the target task.
>
>
>
> **Q3: Given the strong ability of currently emerged language models (e.g. ChatGPT) and the accompanying simpler adaption methods (e.g. in-context learning), the proposed method seems outdated**
>
> R3 : In-context learning has proven to be a highly effective approach. However, large-scale models are sensitive to such hard prompts, making it labor-intensive to design adapted prompts for different tasks.  Additionally, for certain specialized domains or tasks, executing in-context learning may require experts to design highly professional prompts. And prompt tuning automates this process by searching for soft prompts in a continuous space.
>
> Finally, in the revised version, we will use larger-scale backbones for relevant experiments and compare their capabilities with llm's in-context learning."
>
>
>
> **Q4: Do you think the method or idea proposed in this paper can directly facilitate or have any implications on the usage and adaption of cutting-edge large language models (e.g. ChatGPT)?**
>
> R4 : In the ablation study section, we investigated the impact of model scale. On the three datasets we compared, our approach achieved the best performance when scaled up to T5-large. This demonstrates the potential for our method to achieve even greater performance gains when increasing the model scale.
>
> Therefore, I believe that our method is equally applicable to llm. In the revised version, we will also include experiments with larger-scale backbones, such as T5-3B or llama.
>
>
>
> **Q5: How long does TPT need for training and inference and how is this time consumption compared to full-parameter tuning?**
>
> R5 : During training, we may spend slightly more time, but the inference time remains the same. In the PEFT domain, the focus is on the number of parameter adjustments, and since we have fewer adjustable parameters, the computational and storage resources required are significantly reduced.

---

### Official Review · Reviewer_ZdZA · 2023-08-12

**Soundness:** 2

**Excitement:**

2: Mediocre: This paper makes marginal contributions (vs non-contemporaneous work), so I would rather not see it in the conference.

**Missing References:**

He et al., HyperPrompt: Prompt-based Task-Conditioning of Transformers, ICLR 2022

**Paper Topic And Main Contributions:**

This paper proposes two main limitations for prompt tuning: 1) prompt tuning overlooks the inherent differences among instances that may not be suitable for all data instances 2) prompt tuning approaches are hard to be reused or transferred across tasks. In this paper, they suggest token-wise prompt tuning (TPT) by combining token-wise prompt bank with task-specific prompt tuning on multi-task learning scenarios.

**Questions For The Authors:**

See "Reasons to reject".

**Reasons To Accept:**

They introduce token-wise prompt tuning by decomposing soft prompts by constructing a bank of trainable tokens by memory network on multi-task learning scenarios. The results indicate that their proposed model achieves higher performance than baselines on several benchmarks.

**Reasons To Reject:**

1) Lack of baselines

In this paper, they propose prompt-tuning for multi-task learning scenarios. However, there are some missing baselines such as SPT[1], and HyperPrompt[2] that might be strong contenders in multi-task prompt-tuning.

2) Poor Performance

The experimental results show that most improvements from their proposed method are brought by several tasks (e.g. QNLI, SST-2 ...). However, even though they utilize more parameters than other baselines, their TPT represents poor performance than previous works on various tasks(e.g. MNLI, QQP, STS-B ...) in most of their experiments.

3) Lack of Analysis

In this paper, the authors claim that “training additional embedding layer on the target task can provide more appropriate information than a fixed embedding layer, such as SPT”. However, there is no analysis to verify this or prove that their method could actually solve/alleviate the problems. Moreover, the authors did not provide any interpretation for performance degradation on several tasks than other baselines, which occurs in most experiment results.

[1] Bari et al., SPT: Semi-Parametric Prompt Tuning for Multitask Prompted Learning, arXiv 2022

[2] He et al., HyperPrompt: Prompt-based Task-Conditioning of Transformers, ICLR 2022



**Reproducibility:**

4: Could mostly reproduce the results, but there may be some variation because of sample variance or minor variations in their interpretation of the protocol or method.

**Reviewer Confidence:**

4: Quite sure. I tried to check the important points carefully. It's unlikely, though conceivable, that I missed something that should affect my ratings.

---

> ### Author Rebuttal · Authors · 2023-08-27
>
> Thank you for your valuable comments!
>
> **Q1: There are some missing baselines such as SPT, and HyperPrompt that might be strong contenders in multi-task prompt-tuning.**
>
> R1:Our work involves comparing with several strong baseline methods in prompt tuning developed previously, such as ATTEMPT (EMNLP2022) and MPT (ICLR 2023).These approaches utilize the 'T5-base' as backbone, and both the methods proposed by ATTEMPT and MPT are initially trained on six high-resource tasks before adapting to the target task, which are different from SPT.
>
> The method proposed by SPT is first trained on GLUE and then adpted on SuperGLUE, thus only experimental results on SuperGLUE are available, and its performance is inferior to ours => 74.66 (SPT) vs. 77.2 (Ours)
>
> HyperPrompt performs better than us after introducing additional parameters and joint training with the backbone.However, our backbone parameters are frozen, and their training parameters are significantly larger than ours (approximately 3000 times), as shown in the table below:
>
> |             |  Param   | GLUE | SuperGLUE |
> | :---------: | :------: | :--: | :-------: |
> |    Ours     | 0.00035x | 85.6 |   77.2    |
> | HyperPrompt | 1.04000x | 86.8 |   78.9    |
>
> Furthermore, HyperPrompt is more similar to Prefix tuning, requiring the introduction of parameters in front of every layer's K and V, unlike methods like prompt tuning that only need to modify the embedding layer.In situations where the model needs to switch frequently between different tasks, methods like HyperPrompt, which require parameter loading within the model, can result in longer latency.
>
> Finally, in the revised version, we will replicate HyperPrompt's experiment of freezing the backbone and only training additional parameters at the T5-base scale, allowing for a fairer comparison with our TPT method.
>
>
>
> **Q2: Even though they utilize more parameters than other baselines, their TPT represents poor performance than previous works on various tasks in most of their experiments.**
>
> R2:Our method has achieved favorable performance on numerous datasets, and when compared to these baselines, especially full-parameter fine-tuning, TPT has achieved better overall results in both GLUE (84.9 (FT) vs.  85.6 (Ours)) and SuperGLUE (73.9 (FT) vs. 77.2 (Ours)).
>
>
>
> **Q3: The authors claim that “training additional embedding layer on the target task can provide more appropriate information than a fixed embedding layer, such as SPT”. However, there is no analysis to verify this or prove that their method could actually solve/alleviate the problems.**
>
> R3: SPT's performance on SuperGLUE is inferior to ours (74.66 (SPT) vs. 77.2 (Ours)), providing experimental evidence of the superiority of our method.

---

### Meta-Review · Area_Chair_ajTR · 2023-09-15

**Recommendation:** 3

**Metareview:**

This work proposes a new approach for multi-task efficient fine-tuning. Proposed methods creates a prompt bank using a set of "meta-training" tasks and then combines individual items in the prompt bank at evaluation to achieve superior transfer. Reviewers agree on the clarity of writing and presentation and find the proposed method novel. There are some concerns about comparison to relevant work and using of larger models, authors provide comparison to other relevant work and commit to adding larger models by the camera-ready.

---

### Decision · Program_Chairs · 2023-10-07

**Decision:**

Accept-Findings

**Comment:**

This work proposes a new approach for multi-task efficient fine-tuning. Proposed methods creates a prompt bank using a set of "meta-training" tasks and then combines individual items in the prompt bank at evaluation to achieve superior transfer. Reviewers agree on the clarity of writing and presentation and find the proposed method novel. There are some concerns about comparison to relevant work and using of larger models, authors provide comparison to other relevant work and commit to adding larger models by the camera-ready.